# Circulating Hsp70 Levels and the Immunophenotype of Peripheral Blood Lymphocytes as Potential Biomarkers for Advanced Lung Cancer and Therapy Failure after Surgery

**DOI:** 10.3390/biom13050874

**Published:** 2023-05-22

**Authors:** Seyer Safi, Luis Messner, Merten Kliebisch, Linn Eggert, Ceyra Ceylangil, Philipp Lennartz, Benedict Jefferies, Henriette Klein, Moritz Schirren, Michael Dommasch, Dominik Lobinger, Gabriele Multhoff

**Affiliations:** 1Division of Thoracic Surgery, Klinikum rechts der Isar, Technische Universität München (TUM), Ismaningerstr. 22, 81675 Munich, Germany; seyer.safi@mri.tum.de (S.S.); luis.m.messner@gmail.com (L.M.); merten_kliebisch@web.de (M.K.); linneggert@yahoo.de (L.E.); benedict.jefferies@tum.de (B.J.); henriette.klein@mri.tum.de (H.K.); moritz.schirren@tum.de (M.S.); 2Central Institute for Translational Cancer Research Technische Universität München (TranslaTUM), Einsteinstr. 25, 81675 Munich, Germany; ceyra.ceylangil@web.de (C.C.); philipp.lennartz@tum.de (P.L.); 3Emergency Department, Klinikum rechts der Isar, Technische Universität München (TUM), Ismaningerstr. 22, 81675 Munich, Germany; michael.dommasch@mri.tum.de; 4Department of Thoracic Surgery, München Klinik Bogenhausen, Lehrkrankenhaus der Technischen Universität München (TUM), Englschalkinger Str. 77, 81925 Munich, Germany; dominiklobinger@web.de; 5Department of Radiation Oncology, Klinikum rechts der Isar, Technische Universität München (TUM), Ismaningerstr. 22, 81675 Munich, Germany

**Keywords:** lung cancer, surgery, early recurrence, advanced COPD, circulating Hsp70, immunophenotypic profile, biomarker

## Abstract

Lung cancer remains a devastating disease with a poor clinical outcome. A biomarker signature which could distinguish lung cancer from metastatic disease and detect therapeutic failure would significantly improve patient management and allow for individualized, risk-adjusted therapeutic decisions. In this study, circulating Hsp70 levels were measured using ELISA, and the immunophenotype of the peripheral blood lymphocytes were measured using multiparameter flow cytometry, to identify a predictive biomarker signature for lung cancer patients pre- and post-operatively, in patients with lung metastases and in patients with COPD as an inflammatory lung disease. The lowest Hsp70 concentrations were found in the healthy controls followed by the patients with advanced COPD. Hsp70 levels sequentially increased with an advancing tumor stage and metastatic disease. In the early-recurrence patients, Hsp70 levels started to increase within the first three months after surgery, but remained unaltered in the recurrence-free patients. An early recurrence was associated with a significant drop in B cells and an increase in Tregs, whereas the recurrence-free patients had elevated T and NK cell levels. We conclude that circulating Hsp70 concentrations might have the potential to distinguish lung cancer from metastatic disease, and might be able to predict an advanced tumor stage and early recurrence in lung cancer patients. Further studies with larger patient cohorts and longer follow-up periods are needed to validate Hsp70 and immunophenotypic profiles as predictive biomarker signatures.

## 1. Introduction

Lung cancer remains the most frequent cause of cancer-related deaths worldwide [1], with non-small cell lung cancer (NSCLC) accounting for 80–85% of all lung cancer cases. Randomized controlled trials, most notably the National Lung Screening (NLS) and the NELSON trial, have indicated that for high-risk groups, low-dose computed tomography (LDCT) screening has the capacity to detect lung cancer at an earlier stage [2,3,4]. The NLS trial reported a reduction of 20% and 6.7% in lung cancer mortality and all-cause mortality, respectively, with a specificity of 73.4%, whereas the positive predictive value of baseline LDCT screening in the NELSON trial was only 35.7%. A high rate of false-positive lung findings puts patients at risk for complications from unnecessary invasive clinical interventions. Therefore, a high medical need exists for a minimally invasive risk assessment to identify the individuals who will most likely benefit from comprehensive lung cancer screening. However, to date, no validated molecular lung cancer-specific biomarker exists that can increase the positive predictive value and efficiency of the LDCT screening.

Chronic obstructive pulmonary disease (COPD) is a heterogeneous lung condition which shares many risk factors, including smoking, with lung cancer [5,6]. COPD is an independent risk factor for lung cancer and is characterized by similar non-specific symptoms and a similar etiology to NSCLC. Patients with COPD and lung cancer have been reported to share common patterns of the expansion and activation of circulating immune cells, which is associated with an impaired T cell function [7,8]. To distinguish between chronic inflammation caused by COPD and lung cancer, a biomarker signature would be of clinical value.

The major stress-inducible heat shock protein 70 (Hsp70) is a highly conserved molecular chaperone that is ubiquitously expressed in prokaryotic and eukaryotic cells. Physiologically, Hsp70 resides in nearly all subcellular compartments, such as the cytosol, mitochondria, lysosomes and nucleus. Following a large variety of different stress stimuli, the synthesis of Hsp70, which mediates protection against apoptosis and supports the transport and refolding of unfolded proteins and maintains protein homeostasis, is highly upregulated [9]. In contrast to normal cells and in addition to cytosolic localization, tumor cells of different entities present Hsp70 on their plasma membrane and actively release Hsp70 in extracellular micro-vesicles (EVs), such as exosomes [10,11,12] or ectosomes. In a previous study we demonstrated that pre-therapeutic serum Hsp70 levels correlated with the gross tumor volume in 98 patients with NSCLC [13].

EVs containing bioactive proteins, nucleic acids and lipids [14], which are most likely shaped by the physiological state and the EV biogenesis pathway [15], contribute to the intercellular communications that regulate physiological and pathological processes [16]. Recently, novel methods have been established to isolate EVs from the plasma of patients at high purities [17]. HSPA2, a member of the HSP70 family, has been identified as an EV-associated epithelial cell differentiation-related factor [18]. Tumor EVs fulfill a multitude of functions involved in angiogenesis, immunosuppression, invasion and metastatic niche formation, and therefore harbor potential for diagnostic purposes. By using functional single-cell RNA profiling, an EV secretion signature could be identified for triple-negative metastatic breast cancer cells, which are associated with poor survival, invasive tumor growth and immunosuppression [19].

Inflamed and dying cells are known to release Hsp70 into the circulation, but as a free protein rather than in EVs. Therefore, it was assumed that chronic inflammatory conditions in patients with advanced COPD might also cause elevated Hsp70 levels in the blood. In order to evaluate the role of circulating Hsp70 as a tumor-specific biomarker, we included a group of patients with advanced COPD in this study.

Membrane Hsp70 acts as a double-edged sword. On the one hand, a high membrane expression of Hsp70 is associated with increased tumor aggressiveness and a higher risk of metastatic spread [20,21,22]. On the other hand, membrane Hsp70 represents a tumor-specific target for activated natural killer (NK) cells expressing the activatory heterodimeric C-type lectin receptor complex CD94/NKG2C. CD94 on NK cells interacts with membrane Hsp70 on tumor cells, induces granzyme B production and release, and thereby induces tumor cell apoptosis [23,24,25,26].

A better understanding of the role of the immune system in lung cancer could optimize immune-based clinical approaches, such as immune checkpoint inhibitor (ICI) and cell-based therapies. Since CD8^+^ cytotoxic T lymphocytes (CTLs), CD4^+^ T helper cells and CD3^−^/CD56^+^ natural killer (NK) cells are key players in anti-tumor immunity, and the immunosuppressive CD4^+^/CD25^+^/FoxP3^+^ regulatory T cells (Tregs) can contribute to NK and T cell exhaustion [27,28], the question arises as to whether alterations in the composition of these immune cell types in the peripheral blood has any prognostic relevance for the prediction of therapeutic success and the overall survival of patients with lung cancer [28,29].

The aim of this prospective pilot study was to determine the circulating Hsp70 levels in patients with lung cancer pre- and three months post-operatively, in patients with lung metastases, in patients with advanced COPD and in a cohort of healthy individuals, in order to identify the potential of Hsp70 as a biomarker for the presence of lung cancer and for early therapy failure. The hypothesis was that the highest Hsp70 levels are detected in patients with highly aggressive, advanced lung tumors, followed by patients with COPD and the healthy controls. As it is known that CD94^+^ NK cells can be activated by Hsp70 in an inflammatory microenvironment [30], we further speculated that increased proportions of CD94^+^ NK cells in the peripheral blood of patients with lung cancer before surgery is associated with a better, recurrence-free outcome. Therefore, in addition to the circulating Hsp70 levels, the peripheral blood of lung cancer patients was immune profiled using multiparameter flow cytometry, especially with respect to NK cell-related markers.

As expected, the lowest Hsp70 concentrations were found in the healthy controls, followed by patients with advanced COPD. In patients with lung cancer, Hsp70 levels sequentially increased with an advancing stage of disease and with metastatic disease. In lung cancer patients with early recurrence, Hsp70 levels started to increase within the first three months after surgery, but remained unaltered in the recurrence-free patient cohort. An early recurrence was associated with a significant drop in B cells and an increase in the numbers of immunoregulatory T (Treg) cells before and after surgery, whereas recurrence-free patients showed elevated T and NK cell levels before surgery. A more favorable clinical outcome is associated with elevated proportions of cytotoxic T and Hsp70-reactive NK cells before surgery.

## 2. Material and Methods

### 2.1. Patients and Sample Collection

EDTA blood samples (7.5 mL) were prospectively collected between January 2020 and May 2022 from patients with confirmed or suspected lung cancer, with confirmed or suspected lung metastases, with COPD and with benign thoracic surgery conditions, such as pneumothorax, who were treated at the Technical University of Munich, and from healthy volunteers. From the surgical patients blood samples were collected before and three months after surgery. The study was conducted in accordance with the Declaration of Helsinki and approved by the Institutional Review Board (Ethics Committee) of the Medical Faculty of the Klinikum rechts der Isar, Technische Universität München, Munich, Germany (protocol code 2428/09, 27 February 2020).

### 2.2. Therapies

Patients with lung cancer or lung metastases underwent surgery, and COPD patients were hospitalized for non-surgical therapy of concomitant diseases. All surgical patients underwent general anesthesia and endobronchial double-lumen intubation. Surgery was performed through a video-assisted thoracoscopic approach or through a thoracotomy approach. Systematic nodal dissection was performed in all patients with lung cancer and in selected patients with lung metastases. Diagnosis was confirmed by pathologists from the Institute of Pathology, Klinikum rechts der Isar, Technische Universität München.

### 2.3. Measurement of Free and Exosomal Hsp70 in the Blood by Using the compHsp70 ELISA

The compHsp70 ELISA [31] was used to measure the free and exosomal Hsp70 in circulation. Peripheral blood was collected in EDTA KE/9 mL tubes, centrifuged at 1500× *g* for 15 min at 4 °C, and plasma aliquots (300 µL) were stored at −80 °C. Nunc MaxiSorb^TM^ flat-bottom 96 well plates (Thermo Scientific, Rochester, NY, USA) were coated overnight with cmHsp70.2 coating antibody (100 μL) (multimmune GmbH, Munich, Germany) at a dilution of 1 µg/mL in sodium carbonate buffer (0.1 M sodium carbonate (Na_2_CO_3_) pH 11.4, 0.1 M sodium hydrogen carbonate (NaHCO_3_) pH 8.5, Sigma-Aldrich, Saint Louis, MO, USA). Following an incubation period of 12 h, the plate was washed with washing buffer (0.05% *v*/*v* Tween in PBS, 250 μL per well) and blocked for 30 min by adding 250 μL of blocking buffer (Liquid Plate Sealer^TM^, Candor Bioscience GmbH, Wangen im Allgäu, Germany) to prevent nonspecific protein binding. Following another washing step, 100 µL plasma samples (1:5 dilution) in StabilZyme Select Stabilizer (Surmodics, Eden Prairie, MN, USA), as well as an eight-point Hsp70 protein standard (0–100 ng/mL, multimmune GmbH, Munich, Germany) diluted in StabilZyme Select Stabilizer (Diarect GmbH, Freiburg i. Breisgau, Germany), was added to the wells in duplicate. Following an incubation period of 30 min in the dark at room temperature, the plates were washed and incubated with cmHsp70.1-Biotin-labeled detection antibody (100 µL, 200 ng/mL, multimmune GmbH, Munich, Germany) in HRP-Protector (Candor Bioscience GmbH, Wangen i. Allgäu, Germany) for 30 min at room temperature in the dark. After another washing step, horseradish peroxidase (HRP)-conjugated streptavidin (100 µL, 57 ng/mL, Senova GmbH, Weimar, Germany) in HRP-Protector (Candor Bioscience GmbH, Wangen i. Allgäu, Germany) was added for 30 min at room temperature and then the plate was incubated with the substrate reagent (BioFX TMB Super Sensitive One Component HRP Microwell Substrate, Surmodics, Inc., Eden Prairie, MN, USA) for 15 min at room temperature in the dark. The colorimetric reaction was stopped by adding 50 µL of Stop Solution (2 N H_2_SO_4_) and the absorbance at 450 nm and 570 nm was measured in a microplate reader (VICTOR X4 Multilabel Plate Reader, PerkinElmer, Waltham, MA, USA). The absorbance at 450 nm was adjusted by subtracting the absorbance at 570 nm.

### 2.4. Immunophenotyping of Lymphocyte Subpopulations Using Multiparameter Flow Cytometry

The composition of lymphocyte subpopulations in the peripheral blood from the healthy individuals, and the patients with lung cancer before surgery and three months after surgery, was determined using flow cytometry on a FACSCalibur™ flow cytometer (BD Biosciences, Heidelberg, Germany). Briefly, 1.5 mL EDTA blood was diluted in 0.4 mL flow cytometry buffer (PBS containing 10% *v*/*v* heat inactivated fetal calf serum, FCS). For each analysis, 14 aliquots of 100 µL cell suspension were incubated with the fluorescent-labeled monoclonal antibodies shown in Table 1, for 15 min in the dark. After washing with flow cytometry buffer, erythrocytes were lysed by incubating with 2 mL of BD FACS™ Lysing solution (1:9 dilution in double distilled (dd) H_2_O, 349202-BD Biosciences) for 10 min in the dark at room temperature. After another washing step, the cells in tubes 1–11 were analyzed in a FACSCalibur™. The cells in tubes 12–14 were first incubated with the fixation buffer A (1:10 dilution of BD Component A in ddH_2_O) for 10 min, and then with the permeabilization Buffer C (1:50 dilution of BD Component B in Buffer A) for 30 min in the dark at room temperature, together with the fluorescent-labeled antibodies against intracellular antigens expressed by immunoregulatory (Treg) cells, before being analyzed in a FACSCalibur™ flow cytometer.

## 3. Results

### 3.1. Exosomal and Free Hsp70 Levels in Circulation in the Healthy Controls, Different Types of Lung Tumors and Patients with COPD

As shown in the patient flow diagram (Figure 1), out of 65 patients with suspected lung cancer, 49 patients had histopathological-confirmed lung tumors or lung metastases. The patients’ ages at diagnosis ranged from 29 to 82 years with a mean age of 66 years. The circulating Hsp70 levels were measured in the plasma of healthy human volunteers (Healthy, *n* = 108), patients with lung tumors (LT, *n* = 32), lung metastases (METASTASES, *n* = 17) and also in patients with COPD (COPD, *n* = 18), using the compHsp70 ELISA [31]. The characteristics of these patients are summarized in Table 2 and Table 3. Patients with lung cancer were clustered according to their tumor histology, TNM classification and outcome.

A comparison of the circulating Hsp70 levels in the patients with different types of lung cancer and the healthy volunteers revealed significantly elevated Hsp70 concentrations in the patients with adeno carcinoma (ADENO-CA, **** *p* < 0.0001) and lung metastases (MET, * *p* < 0.05) compared to the healthy control cohort (Healthy, *n* = 108) (Figure 2a). The patients with squamous cell carcinoma (SCC) and other types of lung carcinomas (OTLC) also had higher Hsp70 levels compared to the healthy controls, although the values did not reach statistical significance (Figure 2a). Compared to the controls, the COPD patients (COPD, *n* = 18) had significantly higher circulating Hsp70 levels (* *p* < 0.05), albeit these values were significantly lower than those in the adeno carcinoma (ADENO-CA, ** *p* < 0.01), SCC, OTLC and MET patients. These data might provide a first hint that circulating Hsp70 levels can differentiate advanced COPD from lung cancer.

In line with data from a previous study, primary lung tumor patients in stage I (*n* = 7, * *p* < 0.05), stage II (*n* = 5, * *p* < 0.05) and stage III (*n* = 11, ** *p* < 0.01) also had significantly elevated Hsp70 levels (Figure 2b), and the Hsp70 levels gradually increased with a higher tumor stage [23].

### 3.2. Exosomal and Free Hsp70 Levels before and after Surgery in Lung Cancer Patients in Different Tumor Stages

Elevated circulating Hsp70 levels in the patients before surgery are most likely derived from exosomal Hsp70, which is actively released by viable tumor cells, whereas elevated levels among patients after a tumor resection most likely originate from surgery-induced inflammation. A comparison of Hsp70 levels in primary lung tumor patients with tumor stages I–III before (81.3 ng/mL) and three months after surgery (152.7 ng/mL), revealed increased levels compared to the healthy controls (35.06 ng/mL), whereas those of patients with stage IV lung cancer (before surgery 327.5 ng/mL, * *p* < 0.05; after surgery 344.6 ng/mL) and lung metastases (before surgery 295.6 ng/mL; after surgery 282.7 ng/mL, * *p* < 0.05) were even higher (Figure 3).

### 3.3. Exosomal and Free Hsp70 Levels in a Cohort of Recurrence-Free and Early Recurrence Lung Cancer Patients

A total of 16 patients with primary and secondary lung tumors were clustered into two groups based on their clinical outcome twelve months after surgery. Ten patients showed no signs of tumor growth and/or recurrence (*n* = 10) within this follow-up period and were classified as the recurrence-free cohort (RFC). Six patients developed metastases, showed tumor regrowth or recurrence within the first twelve months after surgery (*n* = 6), and were defined as the early recurrence cohort (RC). The clinical characteristics of these patients are summarized in Table 4. The analysis of circulating Hsp70 levels showed significantly elevated levels in the recurrence-free (RFC) patients before surgery (Pre RFC, * *p* < 0.05) but not after surgery, and also in the early recurrence (RC) patients before (Pre RC, * *p* < 0.05) and after surgery (Post RC, ** *p* < 0.01) (Figure 4). The apparent increase in Hsp70 levels three months after surgery in our small early recurrence cohort may underpin the potential of Hsp70 as a potential biomarker for prognosis.

### 3.4. Immunophenotyping of Lymphocyte Subpopulations in a Cohort of Recurrence-Free and Early Recurrence Lung Cancer Patients

In addition to the circulating Hsp70 levels, peripheral-blood-lymphocyte-subpopulation profiling of the recurrence-free (RFC) and recurrence (RC) lung cancer patient cohorts, and of the healthy individuals, was undertaken using multiparameter flow cytometry. A representative example of the lymphocyte gating strategy for the multiparameter flow cytometric analysis of CD3^−^/CD19^+^ B cells, CD3^−^/CD56^+^ NK cells, CD3^+^/CD45^+^ T cells, CD3^+^/CD4^+^ T helper, CD3^+^/CD8^+^ cytotoxic T cells, and CD4^+^ and CD8^+^ CD3^+^/CD25^+^/FoxP3^+^ Treg cells is illustrated in Figure 5.

Overall, the proportion of CD45^+^ lymphocytes was significantly lower in all the recurrence-free (RFC) lung cancer patients before (Pre RFC, **** *p* < 0.0001) and three months after surgery (Post RFC, * *p* < 0.05), and in the cohort of early recurrence (RC) lung cancer patients before surgery (Pre RC; ** *p* < 0.01), as compared to those of the healthy controls. Furthermore, the RC patients after surgery (Post RC) had lower lymphocyte counts, but these values did not reach statistical significance (Figure 6a).

Ionizing irradiation is known to drastically harm B lymphocytes in the circulation of tumor patients, including those with lung cancer [32]. In our recurrence-free (RFC) patient cohort, we were able to demonstrate that surgery did not negatively influence the levels of CD19^+^ B cells in the RFC patient cohort. However, B lymphocytes were significantly decreased in the early recurrence (RC) patient cohort before (Pre RC, * *p* < 0.05) as well as after surgery (Post RC, * *p* < 0.05) (Figure 6b). These data might indicate that the reduction in B cell counts in the peripheral blood is due to the presence of the tumor rather than to the surgical procedure.

The analysis of CD45^+^/CD3^+^ T cells revealed no significant differences between the RFC and RC lung cancer cohorts and the healthy control group (Figure 6c). Although the CD4^+^ T helper cells appeared to be decreased in the RFC patient group post-surgery (Post RFC, ** *p* < 0.01), no change in the CD4^+^ T helper cells was observed in the RC patient cohort before or after surgery. Patients from the RFC cohort before surgery had significantly increased proportions of CD8^+^ cytotoxic T cell lymphocytes (Pre RFC, ** *p* < 0.01) compared to the healthy control cohort (Figure 6c).

The proportion of CD4^+^/CD3^+^/CD25^+^/FoxP3^+^ CD4^+^ Treg cells and CD8^+^/CD3^+^/CD25^+^/FoxP3^+^ regulatory CD8^+^ Treg cells were significantly elevated in all the lung cancer patients before surgery, but not in the RFC group after surgery (Figure 6d). CD4^+^ Treg cells reached their highest levels in early recurrence patients before (Pre RC, ** *p* < 0.01) and also after surgery (Post RC, ** *p* < 0.01), compared to the healthy control cohort. Interestingly, the levels of CD4^+^ Treg cells dropped after surgery in the recurrence-free cohort (RFC), but not in the early recurrence (RC) cohort, indicating that elevated Treg cell counts might be predictive of an unfavorable outcome. Overall, CD8^+^ Treg cells were slightly elevated compared to the healthy cohort, although this difference did not reach statistical significance (Figure 6d).

Concerning the different NK cell subpopulations, it appears that all the receptors on the CD3^−^ NK cells, including the Fc gamma receptor CD16, the activatory C-type lectin receptor NKG2D, the neuronal adhesion molecule CD56, the natural cytotoxicity receptors NKp30 and NKp46 and the Hsp70 interacting receptor CD94, appeared to be elevated in the recurrence-free (RFC) cohort but not in the early recurrence (RC) cohort. However, significant differences were detected only with respect to CD94, the Hsp70 receptor on NK cells, in the RFC (Pre RFC, * *p* < 0.05) group before surgery (Figure 6e). The early recurrence cohort had lower CD94^+^ NK cells before and after surgery (Figure 6e).

## 4. Discussion

In previous studies we demonstrated that circulating Hsp70 levels, which were quantified with the compHsp70 ELISA [31], act as danger-associated molecular patterns (DAMPs) which, in combination with pro-inflammatory cytokines such as IL-2, are able to stimulate membrane Hsp70-specific, NK cell-mediated anti-cancer immune responses in vitro and in vivo, as demonstrated in a clinical phase II trial with NSCLC patients [26,33,34]. Based on these findings, we comparatively investigated the immunophenotype of peripheral blood lymphocytes and the circulating Hsp70 concentrations in lung cancer patients before and three months after surgery. The data from the tumor patients were correlated with the tumor outcome in the time frame of twelve months after surgery, and the blood of healthy volunteers was used as an internal negative control. Due to the relatively small patient cohorts, our results predominantly show trends that, if confirmed by subsequent studies with larger patient cohorts and longer follow-up periods, could confirm our novel immune-related biomarker signature for the prediction of tumor prognosis.

Lung cancer patients and patients with advanced COPD present with similar and rather non-specific symptoms, such as coughing, dyspnea, chest pain or weight loss. Therefore, a biomarker which is able to distinguish between the inflammatory conditions caused by these two diseases would be clinically helpful and beneficial. Most Hsp70 circulating in the blood is derived from Hsp70, which is actively released into extracellular lipid micro-vesicles, such as exosomes, by viable, membrane Hsp70-positive tumor cells [12]. A smaller proportion of free Hsp70 originating from inflamed and dying tumor cells can also be measured in the blood using the compHsp70 ELISA [31]. Since Hsp70 is released by a large variety of different tumor entities, including lung cancer, Hsp70 provides a unique and universal biomarker [13].

In tumor patients the majority of Hsp70 originates from viable tumor cells and, consequently, we hypothesized that extracellular Hsp70 levels might be higher in lung cancer patients than in COPD patients. In this study, we demonstrated that patients with adeno, squamous cell carcinomas, other types of lung carcinomas and lung metastases have elevated Hsp70 plasma levels compared to those from the healthy control group (Figure 2a). Levels of circulating Hsp70 in patients with COPD was also elevated when compared to the healthy controls, but remained significantly lower than in patients with lung adeno carcinoma (Figure 2a), indicating that differences in the circulating Hsp70 levels may be able to distinguish between the two diseases. These findings are in line with the data from Zimmermann et al., who showed that NSCLC patients had higher Hsp70 values than COPD patients [35]. It is important to note that all the COPD patients in our study were hospitalized, either due to an acute exacerbation or another severe non-cancerous disease. Inflammation, stress or traumatic events increase the synthesis of Hsp70 in the cytosol [9,36], which after cell death might transiently increase extracellular Hsp70 levels. Therefore, it is assumed that the elevated Hsp70 levels in patients with advanced COPD might be caused by free Hsp70 released by stressed dying cells. To underline the potential of Hsp70 as a biomarker for COPD, it remains to be determined whether non-hospitalized COPD patients with a lower disease burden also show increased circulating Hsp70 levels. It is expected that those might be lower than those of our severely diseased COPD patients.

In newly diagnosed lung cancer patients, the circulating Hsp70 levels are associated with the tumor stage [30]. Moreover, in this study, patients with metastasized lung cancer in stage IV and patients with lung metastases had higher Hsp70 levels than lung cancer patients in tumor stages I-III (Figure 2b). Previous research has demonstrated that elevated Hsp70 levels reflect the gross tumor volume in NSCLC as determined for stereotactic radiotherapy planning [13]. In this study, circulating Hsp70 levels remained high in patients with lung metastases and in patients with stage IV lung cancer three months after surgery (Figure 3). It should be noted that stage IV lung cancer surgery was only diagnostic in nature and involved patients with disseminated tumors and not those with an oligometastatic disease. A tumor and/or surgery-induced inflammation might be responsible, in part, for the elevated Hsp70 levels at the first follow-up. Future studies with larger patient cohorts and more frequent Hsp70 measurements at later follow-up times after surgery are necessary to evaluate the clinical value of Hsp70 as a prognostic tumor biomarker.

In patients with stage IV lung cancer and lung metastases, Hsp70 levels were significantly higher than in the healthy controls. These data indicate again that higher Hsp70 levels are associated with advanced tumor stages. The fact that Hsp70 levels did not drop three months after the surgical removal of the tumor could be due to the fact that a minimal residual disease in this stage can continue to actively release Hsp70 in exosomes, and/or that surgery-induced inflammation contributes to the elevated free Hsp70 levels.

A comparison of the Hsp70 values with the clinical outcome revealed that patients with a better prognosis (recurrence-free within the first twelve months after surgery) showed no increase in Hsp70 levels three months after surgery, whereas patients who showed tumor recurrence or progression within the first twelve months after surgery showed an increase in circulating Hsp70. These data might indicate that increased Hsp70 levels derived from viable tumor cells might already be able to predict an unfavorable prognosis three months after surgery.

Lymphocytes are the cellular basis of the adaptive [37] and innate immune system and make up 20 to 45% of the leukocytes in the blood circulation [38]. Our data indicate a pronounced lymphopenia in lung cancer patients, which is in line with the finding that a low lymphocyte-to-white blood cell ratio serves as a negative prognostic marker for overall survival [39]. This may be due to the tumor-mediated suppression of the immune system by the release of cytokines, such as tumor growth factor ß (TGF-ß) and interleukin 10 (IL-10); enzymes, such as indoleamine 2,3-dioxygenase (IDO); and inhibitory immune checkpoint ligands, such as programmed death ligand 1 (PD-L1), that can lead to suppressed proliferation, exhaustion and death of the lymphocytes [40].

B cells have been shown to be very sensitive to ionizing irradiation and a dramatic B cell depletion following ionizing radiotherapy (RTx) has been reported for many different tumor entities [41,42]. In the patient cohort examined in this study, only one patient received RTx, whereas seven patients received adjuvant chemotherapy (CTx) after surgery. B cells have also been shown to play a role in the effectiveness of immune checkpoint inhibitor (ICI) therapies [43,44]. Elevated B cell counts impairing T cell-dependent anti-tumor immunity may be associated with a poorer prognosis in patients undergoing PD-1-based immunotherapy [45,46]. Hence, B cell levels combined with tumor histology and staging could play a role in determining a patient’s optimal therapy. A comparison of the B cell counts in the recurrence-free (RFC) and recurrence (RC) patient cohorts (Figure 6b) revealed a significant B cell drop only in the early RC cohort. Therefore, we speculate that surgery with optional adjuvant CTx might be less harmful to the B cells than ionizing irradiation.

As the largest subset of lymphocytes [38], the CD3^+^ T cell population represents the basis for the adaptive immune response. Primed cytotoxic CD8^+^ T cells kill tumor cells or virally-infected cells when foreign antigenic peptides are presented on MHC class I molecules [47]. CD4^+^ T helper cells support the immune response of CD8^+^ T cells and B lymphocytes, and are crucial for initiating and maintaining protective anti-tumor immunity [48]. In our study, all the T cell subpopulations remained relatively unaltered, apart from a significant increase in CD8^+^ cytotoxic T cells in the recurrence-free patient cohort (RFC) before surgery (Figure 6c). The elevated amount of cytotoxic T cells could be responsible for the more favorable outcome of the recurrence-free patients (RFC). Seier et al. showed that a reduction in CD4^+^ T cells correlates with lower plasma levels of pro-inflammatory cytokines, such as IL-2 [30]. Since CD8^+^ T cells are dependent on IL-2 produced by CD4^+^ T cells, as well as the co-stimulation of antigen presenting cells (APCs), we hypothesized that the suppression of CD4^+^ T cells could be one reason why tumors escape CD8^+^ T cell-mediated immunity.

Tumor-mediated immunosuppression could be caused by the release of immunosuppressive cytokines, such as TGF-β and IL-10; the expression of immune checkpoint inhibitors, such as programmed cell death protein 1 (PD-1); or by the presence of immunosuppressive cells, such as CD4^+^ Treg cells [49]. In our study, CD4^+^ Treg cells were found to be significantly elevated in all the lung cancer patients, with the highest Treg cell counts detected in the early recurrence patients before and after surgery (Figure 6d). Due to a simultaneous stimulation by IL-2 and TGF-β, and the expression of the high affinity IL-2 receptor, T cells are more likely to differentiate into CD4^+^ Treg cells that protect the tumor by consuming IL-2 and secreting IL-10 and TGF-β [50,51]. High Treg cell counts are associated with a poor clinical outcome in NSCLC, as immunosuppression may help cancer cells to escape immunosurveillance [52]. Moreover, after surgery CD4^+^ Treg cell levels decreased in the recurrence-free cohort (RFC) but not in the early recurrence cohort (RC). This indicates that the CD4^+^ Treg cell ratio may be associated with tumor progression.

Together with the CD3^+^/CD8^+^ cytotoxic T lymphocytes, CD3^−^ NK cells are considered to be crucial as the first line of defense against cancer without prior antigen stimulation [53]. However, NK cell activity can also be hampered by high levels of CD4^+^ Treg cells secreting anti-inflammatory cytokines (IL-10, TGF-β), by immune checkpoint inhibitors (PD-1, CTLA-4), by a direct cell-to-cell contact or by reduced IL-2 levels [54,55,56,57]. The expression of NK cell receptors, such as the Fc gamma receptor CD16, the C-type lectin receptor CD94 and the activation markers CD69, NKp30, NKp46 and NKG2D, were measured to investigate their anti-tumor potential. The overall increase in all the NK cell receptors, and the significant increase in NK cells expressing the heterodimeric C-type lectin receptor CD94 in the recurrence-free patient cohort (RFC), but not in the early recurrence patient cohort (RC), might indicate that activated NK cells are supporting the anti-tumor immune response.

CD94 in combination with NKG2C is considered to be the activatory receptor complex that enables NK cells to recognize and kill tumor cells expressing Hsp70 on their plasma membrane [58]. Gross et al. showed that CD94 expression on NK cells, as well as cytolytic activity against Hsp70-positive tumor cells, was significantly elevated after a co-incubation of NK cells with Hsp70 and IL-2 [26,58,59]. The high cytolytic activity of NK cells against tumor cells expressing membrane Hsp70, which is associated with an increased expression density of CD94 on NK cells, in the RFC patients before surgery, could be a predictive marker for a more favorable clinical outcome. Moreover, membrane Hsp70-positive tumor cells actively release exosomes that present Hsp70 on their membrane [12] which, in turn, can stimulate the expression of CD94 on NK cells. A subset of CD3^−^/NKG2D^+^ NK cells was slightly decreased in the RC post-surgery. Since an upregulated expression of the activatory receptor NKG2D plays a crucial role in NK cell-mediated immunity against cancer [60], the downregulation of NKG2D in the early recurrence cohort could be a reflection of NK cell exhaustion [61].

## 5. Conclusions

We demonstrated the potential of circulating Hsp70 as a biomarker for lung cancer, as it appears to be able to distinguish between the different stages of lung cancer and might be able to predict early tumor recurrence after complete lung cancer resection with curative intent, as early as three months after surgery. We also speculated that an increased proportion of CD8^+^ T cells and CD94^+^ NK cells, and a decreased proportion of CD4^+^ Treg cells in the peripheral blood might be associated with a more favorable outcome. However, further studies with larger patient cohorts and longer follow-up periods are needed to validate the potential of Hsp70 and the immunophenotypic profile as a predictive biomarker signature for lung cancer screening and monitoring of the therapeutic response.

## 6. Patents

The compHsp70 ELISA is patented by multimmune GmbH, Munich, Germany (US 11,460,472—other applications pending).

## Figures and Tables

**Figure 1 biomolecules-13-00874-f001:**
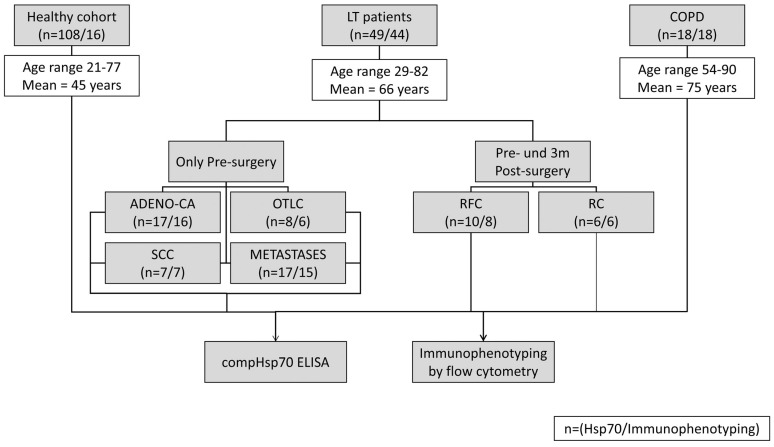
Patient flow diagram of the healthy controls, lung tumor (LT) and advanced-COPD patients. Blood samples were taken from patients with adeno carcinoma (ADENO-CA), squamous cell carcinoma of the lung (SCC), other lung cancers (OTLC) and lung metastases (MET) before (Pre) and three months (3m Post) post-operatively. Patients were divided into a recurrence-free (RFC) and early recurrent (RC) cohort. The numbers in parentheses indicate the number of blood samples which were available for Hsp70 measurements using ELISA and for immunophenotyping using multiparameter flow cytometry.

**Figure 2 biomolecules-13-00874-f002:**
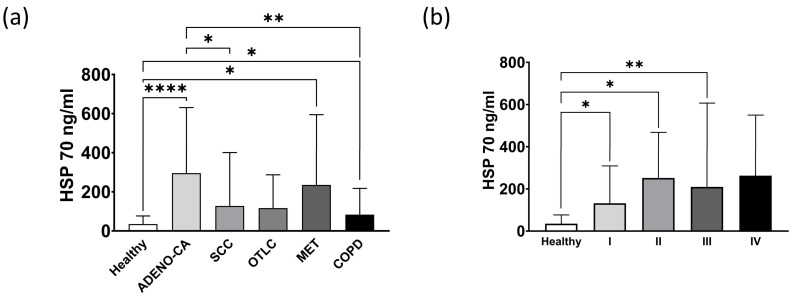
Circulating Hsp70 levels in (**a**) healthy individuals (Healthy, *n* = 108), patients with adeno carcinoma of the lung (ADENO-CA, *n* = 17), squamous cell carcinoma (SCC, *n* = 7) of the lung, other types of lung cancers (OTLC, *n* = 8), lung metastases (MET, *n* = 17) and advanced COPD (*n* = 18). (**b**) Hsp70 levels in the healthy control cohort (*n* = 108) and primary lung tumor patients in stage I (*n* = 7), II (*n* = 5), III (*n* = 11) and IV (*n* = 9). * *p* < 0.05, ** *p* < 0.01, **** *p* < 0.0001.

**Figure 3 biomolecules-13-00874-f003:**
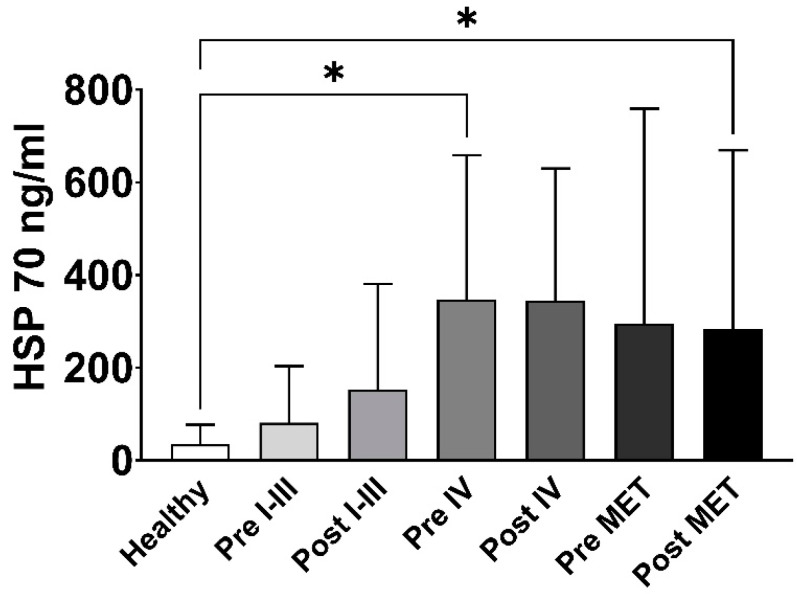
Measurement of Hsp70 in the peripheral blood of healthy individuals (*n* = 108), patients with primary lung tumors in stages I-III (*n* = 6), stage IV (*n* = 3) and lung metastases (MET, *n* = 7), before (Pre) and three months (Post) post-operatively, * *p* < 0.05.

**Figure 4 biomolecules-13-00874-f004:**
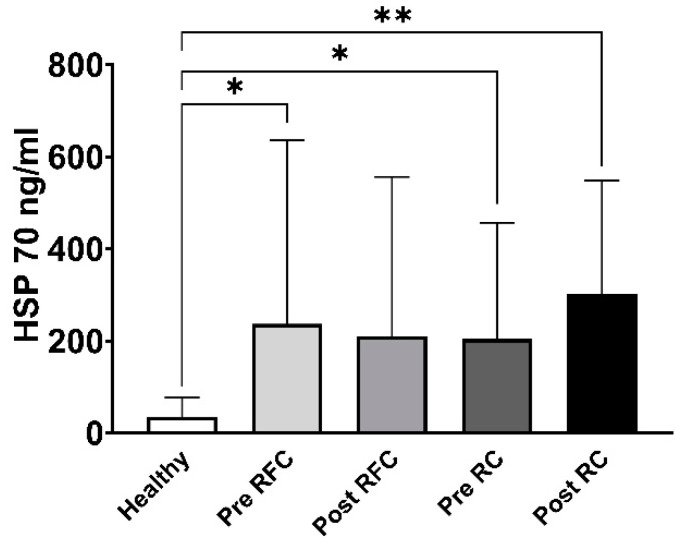
Measurement of Hsp70 levels in the peripheral blood of healthy human volunteers (Healthy, *n* = 108) and lung cancer patients before (Pre) and three months after (Post) surgery in a recurrence-free (RFC, *n* = 10) and early recurrence (RC, *n* = 6) cohort, * *p* < 0.05, ** *p* < 0.01.

**Figure 5 biomolecules-13-00874-f005:**
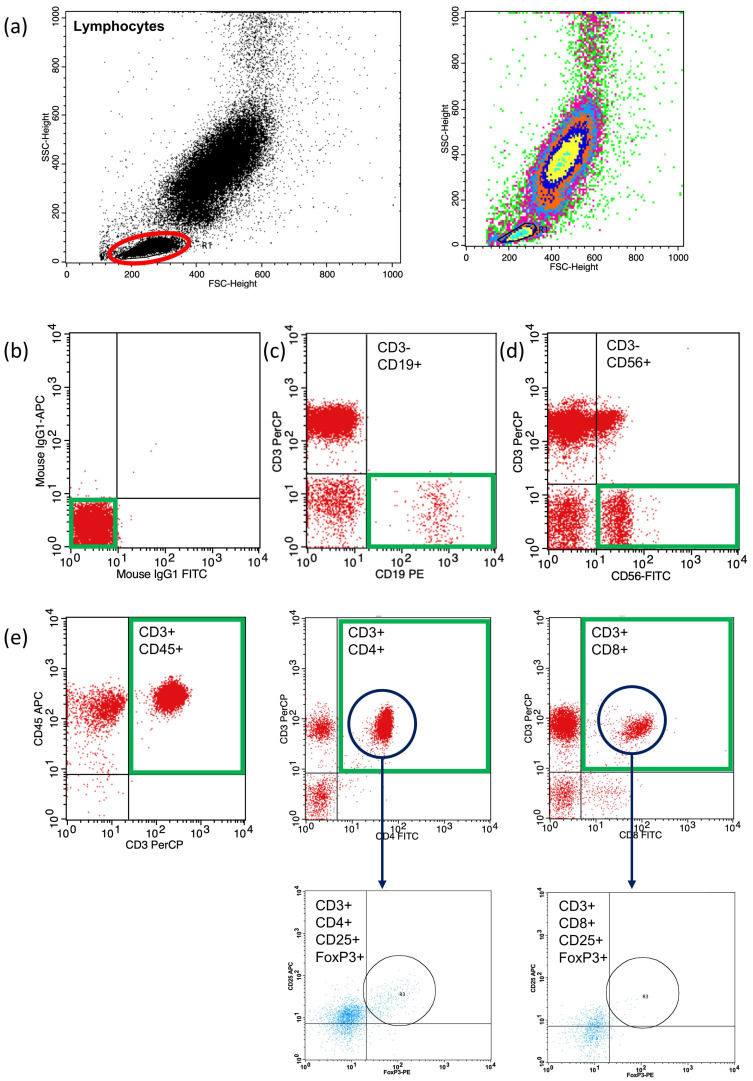
Representative example of a gating strategy to identify lymphocyte subpopulations (**a**) such as CD3^−^/CD19^+^ B cells (**c**), CD3^−^/CD56^+^ NK cells (**d**), CD3^+^/CD45^+^ leukocytes (**e**), CD3^+^/CD4^+^ T helper (**e**), CD3^+^/CD8^+^ cytotoxic T cells (**e**), CD4^+^ and CD8^+^ CD3^+^/CD25^+^/FoxP3^+^ Treg cells (**e**). (**b**) negative control gating, red circle, lymphocyte gate; green box, gated cell population for subpopulation analysis.

**Figure 6 biomolecules-13-00874-f006:**
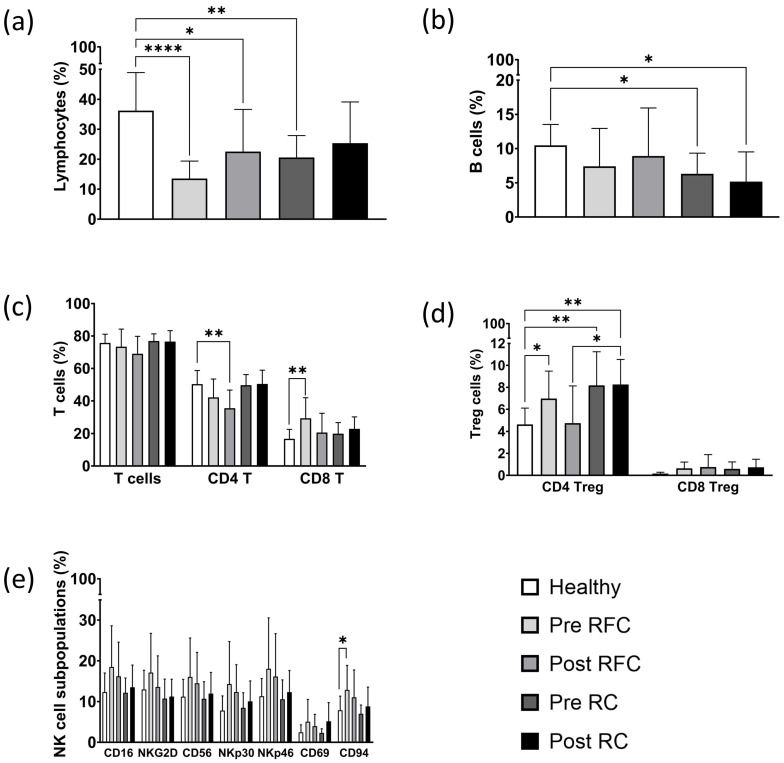
Composition of lymphocytes and lymphocyte subpopulations in the peripheral blood of the healthy volunteers (Healthy, *n* = 16) and in a cohort of recurrence-free (RFC, *n* = 8) and early recurrence (RC, *n* = 6) lung cancer patients before (Pre) and after (Post) surgery, as determined by using multiparameter flow cytometry. (**a**) The proportion of CD45^+^ lymphocytes in the healthy volunteers, pre-surgery recurrence-free (Pre RFC), post-surgery recurrence-free (Post RFC), pre-surgery early recurrence (Pre RC) and post-surgery early recurrence (Post RC) lung cancer patients. (**b**) The proportion of CD19^+^ B cells in all the indicated groups. (**c**) The proportion of CD3^+^ T cells, CD4^+^ T helper cells and CD8^+^ cytotoxic T cells in all the indicated groups. (**d**) The proportion of CD4^+^ and CD8^+^ CD3^+^/CD25^+^/FoxP3^+^ Treg cells in all the indicated groups. (**e**) The proportion of different NK cell subpopulations including CD3^−^/CD16^+^ (CD16), CD3^−^/NKG2D^+^ (NKG2D), CD3^−^/CD56^+^ (CD56), CD3^−^/NKp30^+^ (NKp30), CD3^−^/NKp46^+^ (NKp46), CD3^−^/CD69^+^ (CD69) and CD3^−^/CD94^+^ (CD94) in all the indicated groups (* *p* < 0.05, ** *p* < 0.01, **** *p* < 0.0001).

**Table 1 biomolecules-13-00874-t001:** List of antibody combinations used for immune phenotyping of peripheral blood lymphocytes using flow cytometry.

Tube	Company	Cat. No.	Specificity	Antibody	Volume (µL)
1	BD	345815	Isotype Control	IgG1-FITC	5
	BD	345816		IgG1-PE	5
	BD	345817		IgG1-PerCP	5
	Caltag/Invitrogen	MG105		IgG1-APC	1
2	BD	555888	T/NK cells	CD94-FITC	5
	BD	345812		CD56-PE	5
	BD	345766		CD3-PerCP	10
	Caltag/Invitrogen	MHCD4505		CD45-APC	1
3	BD	345811	T/B/NK cells	CD56-FITC	5
	BD	555413		CD19-PE	20
	BD	345766		CD3-PerCP	10
	Caltag/Invitrogen	MHCD4505		CD45-APC	1
4	BD	345811	T/NK cells	CD56-FITC	5
	BD	555413		CD19-PE	10
	BD	345766		CD3-PerCP	10
	Caltag/Invitrogen	MHCD4505		CD45-APC	1
5	BD	345811	T/NK cells	CD56-FITC	5
	R&D	FAB139P		NKG2D-PE	10
	BD	345766		CD3-PerCP	10
	BD	340560		CD69-APC	5
6	BD	345811	T/NK cells	CD56-FITC	5
	BC	IM3709		Nkp30-PE	10
	BD	345766		CD3-PerCP	10
	BD	340560		CD69-APC	5
7	BD	345811	T/NK cells	CD56-FITC	5
	BC	IM3711		Nkp46-PE	10
	BD	345766		CD3-PerCP	10
	BD	340560		CD69-APC	5
8	BD	555888	T/NK cells	CD94-FITC	5
	R&D	FAB139P		NKG2D-PE	10
	BD	345766		CD3-PerCP	10
	BD	555518		CD56-APC	10
9	BD	555888	T/NK cells	CD94-FITC	5
	BC	IM3709		Nkp30-PE	10
	BD	345766		CD3-PerCP	10
	BD	555518		CD56-APC	10
10	BD	555888	T/NK cells	CD94-FITC	5
	BC	IM3711		Nkp46-PE	10
	BD	345766		CD3-PerCP	10
	BD	555518		CD56-APC	10
11	BD	555346	CD4/CD8 T cells	CD4-FITC	20
	BD	555367		CD8-PE	20
	BD	345766		CD3-PerCP	10
	Caltag/Invitrogen	MHCD4505		CD45-APC	1
12	BD	345815	Isotype Control	IgG1-FITC	5
	BD	345816		IgG1-PE	5
	BD	345817		IgG1-PerCP	5
	Caltag/Invitrogen	MG105		IgG1-APC	1
13	BD	555346	CD4 Treg cells	CD4-FITC	20
	BD	345766		CD3-PerCP	10
	BD	340907		CD25-APC	5
	BD	560046		FoxP3-PE	20
14	BD	555366	CD8 Treg cells	CD8-FITC	20
	BD	345766		CD3-PerCP	10
	BD	340907		CD25-APC	5
	BD	560046		FoxP3-PE	20

Abbreviations: BD—Becton Dickinson, BC—Beckmann Coulter.

**Table 2 biomolecules-13-00874-t002:** Lung cancer patient cohort before surgery for Hsp70 measurements: Sex, age, histology and UICC staging (TNM 8th edition).

Characteristic	Clusters	*n*
Sex	Male	33
	Female	16
Age	20–29	1
	30–39	2
	40–49	2
	50–59	6
	60–69	15
	70–79	16
	80–89	7
Histology	ADENO-CA *	17
	SCC *	7
	OTLC *	9
	MET *	16
UICC stage	I	7
	II	5
	III	11
	IV	9

* Adeno carcinoma (ADENO-CA), squamous cell carcinoma (SCC), other types of lung cancers (OTLC) and lung metastases (MET).

**Table 3 biomolecules-13-00874-t003:** COPD patients: sex, age, COPD GOLD and exacerbation status.

Characteristic	Clusters	*n*
Sex	Male	11
	Female	7
Age	50–59	2
	60–69	3
	70–79	5
	80–89	7
	90–99	1
COPD GOLD	I	0
	II	4
	III	9
	IV	5
Exacerbation	Yes	10
	No	8

**Table 4 biomolecules-13-00874-t004:** Lung cancer patient cohort before and after surgery: Sex, age, histology and UICC staging (TNM 8th edition).

Characteristic	Clusters	*n*
Sex	Male	10
	Female	6
Age	20–29	0
	30–39	2
	40–49	1
	50–59	3
	60–69	6
	70–79	3
	80–89	1
Histology	ADENO-CA *	4
	SCC *	1
	OTLC *	4
	MET *	7
UICC stage	I	1
	II	1
	III	4
	IV	3

* Adeno carcinoma (ADENO-CA), squamous cell carcinoma (SCC), other types of lung cancers (OTLC) and lung metastases (MET).

## Data Availability

The data presented in this study are available upon request from the corresponding author.

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
