# Peer review of "Circulating Hsp70 Levels and the Immunophenotype of Peripheral Blood Lymphocytes as Potential Biomarkers for Advanced Lung Cancer and Therapy Failure after Surgery"

_biomolecules, 2023, doi:10.3390/biom13050874_

Round 1

Reviewer 1 Report

In this excellent paper the authors not only expanded further their idea about the biological role of circulating Hsp70 levels, but also provided evidences on its use as a predictive biomarker signature. As they outlined, lung cancer remains a devastating disease with a quite poor clinical outcome. Accordingly, the central question of the current investigations is the identification of a proper biomarker signature whith the capacity of distinguishing between advanced COPD from lung cancer. Moreover, they also addressed the ability of such a biomarker signature to detect therapeutic failure of lung cancer in order to improve patient management and individualized, risk-adjusted therapeutic decisions.

As they concluded circulating Hsp70 concentrations might have the potential to distinguish COPD from lung cancer and might predict advanced tumour stages and early recurrence in lung cancer patients. I felt especially convincing, that a comparison of the circulating Hsp70 values with clinical outcome explored, that in patients with a better prognosis showed indeed no increase in Hsp70 levels three months after surgery. By contrary,  patients who showed tumour recurrence or progression within the first twelve months explored an increase in the level of their circulating Hsp70. Taken together, these data might indicate that indeed, increased circulating Hsp70 levels, derived mostly from viable tumour cells, are able to predict an unfavorable prognosis. As the authors also outlined, additional studies with larger patient cohorts and longer follow-up periods are necessary to validate the concept outlined here about Hsp70 (and immunophenotypic profiles) as a predictive biomarker signatures.

The paper is clearly written and the evidences justify the conclusions. All the experimental protocols are given in sufficient detail. In my opinion, this is an excellent paper and is of sufficient general interest to warrant publication.

Minor comment: as the authors outlined,  majority of Hsp70 circulating in the blood is most likely derived from the distinct Hsp70 pool, i.e. which is actively released in extracellular lipid microvesicles such as exosomes by viable, membrane Hsp70 positive tumour cells, whereas a smaller proportion of free Hsp70 is likely originating from dying tumour cells. To make the paper more accessible to a diverse audience, I would recommend to add a few more sentences (and references) about the present concepts of extracellular vesicle formation in normal vs tumour cells. For example, several lipids and lipid metabolizing enzymes have been involved in formation and release of exosomes. Lipids are key players of the plasma membrane localization of Hsp70. Based on the current focus on the potential clinical applications of exosomes, not only their cargo (Hsp70, etc.) but also their lipid composition (the mechanism of selection of lipid species into tumor exosomes) would be an interesting topic to be discussed. Emerging single-cell technologies provide a new opportunity to profile individual cells within tumours and investigate what roles they play in these processes.

Author Response

The authors want to thank the reviewer for constructive suggestions, please find enclosed a point by point letter to both reviewers asan attachment.

Reviewer 2 Report

In this manuscript, the authors evaluated the potentiality of circulating Hsp70 levels as a biomarker for the presence of lung cancer and therapy failures. The authors included in the study patients with lung cancer pre- and three months post-operatively, patients with lung metastases, patients with advanced COPD, and healthy individuals.

I found the topic of your manuscript interesting as also the experimental approach that was used and the results obtained. My biggest concern is the comparison between patients with lung cancer and patients with COPD. Lung cancer diagnosis is performed by chest x-ray, CT, MRI… and it is clearly distinguishable from COPD and treated in very different ways, despite the similar non-specific symptoms described. Therefore, in my point of view, a comparison only between lung cancer and metastatic lung cancer or pre and post-surgery could be more interesting as well as the evaluation of Hsp70 as a biomarker of therapy failure after surgery.  I don’t see the point in evaluating the levels of a single protein comparing blood samples from COPD and lung cancer 

Author Response

The authors want to thank the reviewer for helpful suggestions. Please find a point by point letter to both reviewers as an attachment.

Round 2

Reviewer 2 Report

I accept the authors' response